# Prevention of Type 1 Diabetes: Past Experiences and Future Opportunities

**DOI:** 10.3390/jcm9092805

**Published:** 2020-08-30

**Authors:** Przemysław Beik, Martyna Ciesielska, Maria Kucza, Alicja Kurczewska, Joanna Kuźmińska, Bartosz Maćkowiak, Elżbieta Niechciał

**Affiliations:** Department of Pediatric Diabetes and Obesity, Poznan University of Medical Sciences, Szpitalna Street 27/33, 60-572 Poznan, Poland; beikprzemek@gmail.com (P.B.); martyna.ciesielska99@gmail.com (M.C.); kucza.maria@gmail.com (M.K.); alka.kurczewska@gmail.com (A.K.); joasia333444555@wp.pl (J.K.); barmackowiak@gmail.com (B.M.)

**Keywords:** type 1 diabetes, autoimmune diseases, prevention, beta-cell function, clinical trials

## Abstract

Type 1 diabetes (T1D) results from autoimmune destruction of insulin-producing beta-cells in the pancreas, caused by the interplay of genetic and environmental factors. Despite the introduction of advanced technologies for diabetes management, most patients fail to achieve target glycemic control, and T1D still has a high burden of long-term end-organ complications. Over several decades, multiple clinical trials have attempted to find prevention for T1D in at-risk individuals or to stabilize, ultimately reverse, the disease in those with T1D. To date, T1D remains yet incurable condition; however, recently improved understanding of the natural history of the disease may lead to new strategies to preserve or improve beta-cell function in those at increased risk and T1D patients. This publication aims to provide an overview of past experiences and recent findings in the prevention of T1D.

## 1. Introduction

Type 1 diabetes (T1D) is one of the most common chronic diseases affecting children, adolescents, and young adults at crucial times of growth and development [1]. The disease results from a complex interplay between genetic, environmental, and immune factors that initiate T cell-mediated destruction of pancreatic beta-cell [2,3]. The main T1D susceptibility locus maps to the class II loci human leukocyte antigen (HLA), HLA-DRB1, HLA-DQB1, and HLA-DQA1 and contributes up to 30–50% of genetic T1D risk [2]. Table 1 summarizes HLA risk haplotypes in populations of different ethnic backgrounds. Numerous putative environmental factors triggering autoimmunity, such as diet, vitamin D intake, infections, and gut microbiota, may play a role in favoring T1D development [3]. Both the immune regulation and the immune response contribute to T1D pathogenesis, in which cellular immunity plays a vital role [4]. Biomarkers of the immune destruction of the beta-cell include islet-cell antibodies (ICA), glutamic acid decarboxylase autoantibodies (GAD-ab), protein tyrosine phosphatase autoantibodies (IA2-ab), insulin autoantibodies (IAA), and autoantibodies against zinc transporter 8 (ZnT8) [5,6].

Despite the introduction of advanced technologies for diabetes management, individuals with T1D may experience a reduced life expectancy; consequently, researchers continue to seek methods not only to prevent T1D, but also to halt progression or even reverse the disease [10]. Up to date, prevention trials had mixed results, showing promise for improving in beta-cell function, while others have failed to stop the autoimmune process. Recently enhanced understanding of the natural history of the disease helped identify earlier pre-symptomatic phases of T1D, from the asymptomatic stage to clinical diagnosis. These include stage 0, in which individuals carrying T1D susceptibility alleles have not yet developed islet autoantibodies. Stage 1 is characterized by the appearance of multiple islet antibodies with normoglycemia, while stage 2 shows a progression to dysglycemia, and stage 3 is the onset of symptomatic diabetes [11]. The increase in understanding of the natural history of T1D has enabled the use of preventive therapies to delay and ultimately to prevent symptomatic disease at the earliest stages. Approaches to prevent T1D include avoidance of environmental triggers of islet autoimmunity, antigen-specific “vaccination” using islet autoantigens, non–antigen-specific systemic therapies, including immunosuppression or immunomodulation, and cellular therapies. This article provides an overview of past experiences and recent findings in preventing the pre-symptomatic and symptomatic stages of T1D. Figure 1 summarizes the critical stages in the development of T1D and prevention opportunities.

## 2. Primary Prevention

The target population for primary prevention trials is individuals who carry high-risk genotypes before the first appearance of islet autoantibodies. These trials, summarized in Table 2, commonly include mainly low-risk dietary intervention: The avoidance of cow’s milk or gluten, and supplementation of n-3 fatty acids or vitamin D.

TRIGR (Trial to Reduce IDDM in the Genetically at Risk) study is an international trial that was testing effects cow’s milk in genetically at-risk infants with a positive family history for T1D. In the TRIGR study, newborns were randomly assigned to feeding up to 6–8 months of age with a conventional cow’s milk-based formula or a casein hydrolysate formula. Then, participants were followed up for ten years. The cumulative risk of T1D was 8.4% in those receiving casein hydrolysate compare to 7.6% among those on the regular formula. The TRIGR study indicated that a hydrolyzed casein formula did not prevent from developing T1D in genetically at-risk children, and did not delay the onset of the disease [14].

Likewise, the Finnish Dietary Intervention Trial for the Prevention of T1D (FINDIA), was designed to determine whether a formula free of bovine insulin may reduce beta-cell autoimmunity. FINDIA study included newborns having high-risk HLA randomized to cow’s milk formula, whey-based hydrolyzed formula, or whey-based formula free of bovine insulin during the first six months of life whenever breast milk was not available. Participants were followed until the age of three years for the development of T1D-related autoantibodies. The study showed that children receiving the bovine insulin-free had a reduced risk of developing one islet autoantibody compared to children on other formulas [15].

In turn, the BABYDIET study was designed to establish whether delayed exposure to gluten in genetically at-risk infants is feasible, and may prevent islet autoimmunity. Newborns were randomized to the introduction of gluten at 6 or 12 months and followed-up every three months to 36 months of age. This study concluded that late exposure to gluten did not decrease the risk of islet autoimmunity or the development of T1D in genetically at-risk children [16].

The docosahexaenoic acid (DHA) hypothesis was tested by the Type 1 Diabetes TrialNet study group. This pilot study evaluated an anti-inflammatory effect of omega-3 fatty acid supplementation with DHA on islet autoimmunity development. Eligible to participate were pregnant mothers after the 24th week of pregnancy and their babies during the first five months of life, both having a genetic risk for T1D. Participants received either DHA or a placebo. In this study, no effect on autoimmunity has been observed [17].

Vitamin D supplementation in early life has attracted attention as playing a possible role in reducing the risk of T1D in later life. Nevertheless, the prospective Diabetes Autoimmunity Study in the Young (DAISY) had shown that the intake of vitamin D during childhood was not associated with decreased risk of the autoimmune responses [18].

Thus far, none of the specific dietary factors has been proved to be a definite risk factor halting or inducing beta-cell autoimmunity, and their effects have still been contradictory. More recently, primary intervention trials have started focusing on the modulation of the immune system by antigen-specific immunotherapy (ASI) [21]. The goal of ASI is to induce tolerance to known autoantigens to effectively control the autoimmune response via induction or restoration of beta-cell specific tolerance. In the non-obese diabetic (NOD) mouse, the administration of insulin via a variety of routes has shown efficacy in preventing T1D [22]. Based on this finding, oral insulin administration was also tested in children having high-risk HLA in a dose-finding and safety study, the Pre-POINT (Primary Oral/Intranasal Insulin Trial). This pilot study demonstrated that oral administration of 67.5 mg of insulin, compared with placebo, results in an immune response without hypoglycemia [19]. After having demonstrated that mucosal administration of insulin is safe, the GPPAD-POInT (Global Platform for the Prevention of Autoimmune Diabetes Primary Oral Insulin Trial) study was initiated to test whether daily administration of oral insulin, from age 4.0 months to 7.0 months until age 36.0 months to children who are genetically at risk will reduce the beta-cell autoantibodies and T1D development. Infants are randomly assigned to receive an increasing dose of insulin (7.5–67.5 mg/day) or a placebo, and they will be followed for a maximum of seven years. The first participant was enrolled in the GPPAD-POInT in February 2018 [20].

The long-term observations have suggested that certain viral infections may promote the autoimmune attack directed to pancreatic beta-cell in genetically susceptible individuals [23,24,25]. Numerous viruses, including rotavirus, adenovirus, cytomegalovirus, mumps virus, Epstein-Barr virus, or rubella virus, were hypothesized to play a role in the pathogenesis of T1D [24,26]. However, the prime viral candidates for causing T1D are enteroviruses, particularly the Coxsackie B (CVB). Even though the role of enteroviruses in the development of T1D has been investigated for many years, the knowledge of the way enteroviruses act on beta-cell is scarce [27]. It is proposed that viral-induced autoimmunity is activated via several mechanisms, such as epitope spreading, bystander activation, molecular mimicry, and immortalization of infected B cells [26]. The link between T1D and CVB infection has prompted efforts to develop vaccines targeting CVBs. Recently, researchers have produced a novel hexavalent vaccine that protects against the six known CVB serotypes. The vaccine was investigated in different animal models and was shown to protect mice infected with CVB from developing virus-induced T1D. Then, the vaccine was tested in nonhuman primates that have very similar genetics to humans. In these animals, the vaccine-induced antibodies to CVB, suggesting it could protect against the virus. The clinical studies testing the vaccine against CVB in human subjects are on the horizon [28]. The CVB vaccine could be effective for the primary prevention of T1D by halting an induction of beta-cell autoimmunity.

## 3. Secondary Prevention

Secondary prevention is aimed to slow or halt the progressive beta-cell destruction in asymptomatic individuals with persistent islet autoantibodies (stage 1 and 2 of T1D). The approach of secondary prevention trials, shown in Table 3, mostly involves the use of nicotinamide, ASI, immunotherapy with monoclonal antibodies, immunosuppressive drugs, hydroxychloroquine, or anti-inflammatory agents.

Nicotinamide, a water-soluble vitamin (B6) isolated from nicotinic acid, is effective in preventing T1D in animal models [44]. On this basis, several studies, including The Deutsche Nicotinamide Intervention Study (DENIS) and European Nicotinamide Diabetes Intervention Trial (ENDIT), were designed to evaluate the clinical efficacy of high doses of nicotinamide (1.2 g/m^2^) in children having the risk for T1D. The outcome of these two trials showed that the intervention had no effect on halting on preventing the clinical onset of the disease in autoantibody-positive individuals [29,30].

Another strategy of secondary prevention is ASI using insulin as the target antigen. The first clinical trial tested whether intervention with insulin can delay the appearance of overt diabetes was the Diabetes Prevention Trial–Type 1 (DPT-1). The study had two arms, based on the estimated risk of developing T1D within five years. Estimations of risk were based on ICA titers, first-phase insulin response, and the lack of known protective HLA allele. The high-risk group participants (first arm) were randomly assigned to an intervention of low-dose subcutaneous insulin (0.25 U/kg/day given twice a day plus annual 4-day iv insulin infusions) or close observation. The result failed to demonstrate that parenteral insulin can delay or prevent T1D [31]. In this arm, a low-dose of insulin was examined, which may not be enough for the immune-modulating protective effects in high-risk individuals. Therefore, DPT-1 also included a second arm testing whether oral insulin at a higher dose (7.5 mg/a day) can delay T1D development in the intermediate-risk group. Overall, the study did not show a benefit of oral insulin [32]. However, in *a post hoc* analysis of individuals with high titer IAA, oral insulin appeared to delay progression to T1D by about four years [45]. Despite unsatisfactory therapeutic effects, the DPT-1 study has provided an important of data concerning prevention and trial designed. In October 2001, the DPT-1 centers expanded and evolved to the Type 1 Diabetes TrialNet Consortium seeking to prevent or delay the development of T1D in individuals at risk. This led to the initiation of another oral insulin prevention study (TrialNet Study), further investigating *the post hoc* finding of the second arm of DPT-1 study. Again, this trial failed to hit the primary endpoint of slowing or halting T1D onset. However, it should be mentioned four different cohorts of participants enrolled in this trial according to autoantibodies profiles and first-phase insulin release. In the group with lower insulin production and confirmed IAA in addition to GAD/IA-2, oral insulin delayed T1D onset by an average of 31 months [33]. Similar to the DPT-1 trial, a certain group of participants responded.

In addition to oral insulin, it was also hypothesized that nasal insulin might induce immune tolerance. Hence, several studies have been performed to test the effects of nasal insulin on T1D prevention. The first to be reported was a pilot crossover study examining antibody-positive individuals, the Intranasal Insulin Trial (INIT). Participants were randomized to receive intranasal insulin (1.6 mg) spray or placebo daily for ten days and then two days per week for six months, before crossover. INIT revealed no significant effects on beta-cell function; nevertheless, the immune tolerance to insulin was improved [34]. This finding justified the development of a formal trial (INIT-II) to examine whether intranasal insulin (either 1.6 mg or 16 mg) is effective in children and young adults with a risk of progression to T1D. The study included 12 months of intervention and a further four years of observation. During this period, 38% of participants developed T1D, and the younger age was associated with more rapid progression to T1D. Overall, the trial showed that intranasal insulin was safe and induced immune response, but then again did not affect the prevention of T1D [35].

To date, screening studies have mainly included first-degree relatives of T1D patients. Nevertheless, around 85% of new-onset patients have no family history of T1D, so to have a major impact, a preventive effort will require the inclusion of the general population. Planning intervention for this group is challenging and involves an expensive screening strategy. The Fr1da-/Fr1da-Plus-Study, for instance, was designed to screen the general population for early-stage T1D. Islet autoantibodies screening is performed in the pediatric primary care setting [46]. Then, participants diagnosed with pre-symptomatic T1D are invited to participate in the second step of the project, which is the Fr1da Insulin Intervention Study. This trial aims to test whether daily administration of up to 67.5 mg oral insulin for 12 months to children with at least two islet autoantibodies will have the potential to control the autoimmune responses by diverting the immune system to a protective rather than destructive response. The study is currently recruiting participants [36].

The other, the Diabetes Prediction and Prevention (DIPP) Study performed to investigate the potential effect of nasal insulin on reducing the incidence of T1D. DIPP cohort included newborns at high-risk from the general population who were randomized to receive insulin or placebo with a further ten years of follow-up. The study failed to prove the protective role of nasal insulin on disease progression [37]. Although there is no direct benefit, described studies have demonstrated that insulin is safe and could induce immune tolerance. Further studies should focus on finding an appropriate dose of insulin, as well as, should consider the possibility that the immune response to autoantigen may be related to the HLA-DQ genotype of the individuals, because the analysis of insulin alone might not be sufficient to obtain desired outcomes.

Another diabetes-related antigen able to solicit immune response is glutamic acid decarboxylase (GAD). The GAD enzyme, widely distributed in neuroendocrine tissues, transforms glutamate to GABA (gamma-aminobutyric acid). GAD/GABA is also detected in certain non-neural cells, such as pancreatic islet beta-cell, where might regulate hormone release in the pancreas and/or work as a paracrine signaling molecule between the endocrine cells of the islets [47]. Although its role is presently unclear, GAD seems to be one of the most important pancreatic islet beta-cell autoantigens, and autoantibodies toward GAD predict the development of T1D. In NOD mice, the administration of GAD delays or prevents insulitis and T1D [48]. Given that, GAD is being tested as a potential ASI in humans [38].

Diabetes Prevention-Immune Tolerance (DiAPREV-IT) tested the safety and efficacy of ASI with GAD–Alum in non-diabetic individuals with multiple islet autoantibodies. The study cohort was randomly assigned to 2 injections of 20 μg GAD-Alum or placebo, 30 days apart. The participants were observed for five years. The DiAPREV-IT has proved that GAD-Alum is safe in children; however, it did not affect progression to T1D [38]. Despite the DiAPREV-IT failed, a second trial (DiAPREV-IT2) testing GAD-Alum (20 µg twice in 30 days), in combination with high-dose vitamin D3 (2000 U/daily) in children with multiple islet autoantibodies is ongoing [39]. The results are awaited.

Immunomodulation with autoantigens could still potentially prevent T1D. Thus far, the potential mechanism of ASI using GAD-Alum is unknown. However, more recently, it has been shown that the T cell responses induced upon immunization with GAD-Alum are not only directed to T helper (Th) cells two but also Th1. Activation of a distinctive bifunctional phenotype may explain a lack of clinical efficacy of GAD-Alum vaccination. This outcome might be important in understanding therapeutic responses [49]. Further studies are needed to explore the mechanism of action of ASI via GAD-Alum.

Immunotherapy with monoclonal antibodies has diverse results; however, this therapy holds a promise, with at least transient improvement in beta-cell function compared with randomized control groups. The anti-CD3 monoclonal antibody (teplizumab) has attracted considerable attention, and it is one of the most widely investigated immunological approaches to T1D. Teplizumab is targeting CD8 lymphocytes, key effectors of beta-cell injury in T1D pathogenesis. Nevertheless, the mechanism of action of this drug is not completely understood. Recently, the Type 1 Diabetes TrialNet Study Group has tested whether teplizumab treatment would prevent or delay the onset of clinical T1D in individuals at high risk. Participants were randomly assigned to a single 14-day course of teplizumab or placebo, and follow-up for progression to clinical T1D. Although the trial had limitations and the cohort was small, this is the first study showing that treatment with teplizumab can delay T1D diagnosis a median of 2 years in high-risk participants [40].

TrialNet is currently conducting the other two studies testing immune therapies to delay or prevent T1D in individuals in stage 1 T1D. The first, Abatacept Prevention Study, is exploring the use of an immunosuppressive drug as a potential approach to slow the progression of the disease. Abatacept (CTLA-4 Ig) is an immunomodulatory molecule-blocking T cell co-stimulation modulator, having an impact on the rate of reduction of beta-cell function. CTLA4-I is proposed to regulate, but not delete, T lymphocytes through inhibiting their stimulatory pathway of activation; therefore, it is considered relatively safer than other immunosuppressive agents. The primary endpoint will determine whether Abatacept administration will prevent further progression and T1D diagnosis [41]. At the same time, the second trial is Hydroxychloroquine (HCQ) Prevention Study testing for the first time a possible role of HCQ in preventing T1D in individuals with multiple islet autoantibodies and normal glucose tolerance. HCQ has not previously been studied as a treatment to prevent T1D; however, it is already used to decrease the progression of other autoimmune diseases, such as rheumatoid arthritis and lupus. The primary outcome will be to abnormal glucose tolerance (Stage 2) or T1D (Stage 3) [42].

Finally, the Study to Evaluate SIMPONI (Golimumab) is testing whether golimumab treatment is safe and well-tolerated in participants with stage 2 T1D [43]. Golimumab, a tumor necrosis factor-alpha (TNFa) inhibitor, is used to treat rheumatic diseases. TNFa is a cytokine involved in the acute inflammatory process; hence, it may play a role in the pathogenesis of beta-cell destruction in T1D [50,51]. The reduction of TNFa levels has been revealed to ameliorate symptoms of diseases characterized by inflammation, due to autoimmune or hyperimmune reactions [51]. If golimumab appears to be safe, it could find a place in T1D prevention. Further studies are crucial to explore this hypothesis.

## 4. Tertiary Prevention

Tertiary prevention is aimed at new-onset patients, at which time the majority of the beta-cell has already been lost. Then, the purpose of these trials is to preserve the remaining islet beta-cell to induce and prolong partial remission. Several immunomodulatory and immunosuppressive agents have been investigated, alone or in combination, to halt the destructive autoimmune process of beta-cell that occurs in T1D. So far, with varying success to preserve residual beta-cell function. Tertiary prevention trials are summarized in Table 4.

Azathioprine is an immunosuppressive drug that inhibits T cell responses to antigens. The historical study of patients treated with azathioprine and glucocorticoids showed that half of the treated participants become insulin-free, as compared with only 15% of patients in the placebo group [52]. However, only three patients receiving treatment remained in remission at one year. Another non-antigen-specific immunomodulator, cyclosporine A, was proved to be effective in prolonging insulin production. However, the beneficial effect was lost after discontinuing treatment, and almost all patients required insulin again within three years [53]. Although cyclosporine had a transient effect, the study showed a potential role of immunosuppression in slowing the destruction of the beta-cell. Moreover, responses to cyclosporine might depend on autoantibodies profile at diagnosis, and a Canadian-European cyclosporine study showed that those without IA2-ab at randomization had better beta-cell function than IA-2-positive participants [76].

Recently, two monoclonal CD3 antibodies have been widely studied, otelixizumab, and teplizumab, in newly diagnosed T1D patients [54,55,57,58,77]. The Phase II trial with otelixizumab, a humanized non-mitogenic CD3, appeared to suppress the rise in insulin requirements over 48 months; however, the effect was related to age and residual c-peptide at T1D onset. Moreover, the administration of anti-CD3 antibody was associated with significant but transient adverse effects, such as fever or reactivation of Epstein-Barr virus (EBV) [54,55]. The subsequent Phase III trial durable-response therapy evaluation for early or new-onset type 1 diabetes—DEFEND-1 and -2, with a goal of reducing rates of EBV reactivation, used a cumulated dose of 3 mg as compared to 8 mg for the phase II study. At a low dose, EBV reactivation rates were insignificant, but there was no preservation of beta-cell [56].

The Protégé study has examined regimens of teplizumab in new-onset T1D patients. Participants were randomized to four groups—14-day full dose, 14-day low dose, 6-day full dose, or 14-day placebo at baseline and six months. The study did not meet primary endpoints for insulin usage after 1 year (0.5 unit/kg/day) with HbA1c (glycated hemoglobin) < 6.5%. Nevertheless, in the 14-day high dose subgroup, significant efficacy of the treatment was observed after a 2-year follow-up with improved c-peptide responses. The effect was related to younger age, disease duration <6 weeks, HbA1c < 7.5%, insulin dose of < 0.4 U/kg/day, residual c-peptide at onset [57]. These findings suggest that teplizumab is most effective in preserving c-peptide secretion when administered early in the course of the disease.

The AbATE (Autoimmunity-Blocking Antibody for Tolerance in Recently Diagnosed Type 1 Diabetes) group also investigated teplizumab in an unblinded trial in newly diagnosed T1D patients. Individuals were randomly assigned to teplizumab (14-day course at baseline and one year) or no therapy. After two years, the treatment group had significantly better preservation of c-peptide compared to the control group. There was no significant difference in HbA1c between the groups during the study period, but patients receiving teplizumab required less insulin to achieve similar metabolic control. This benefit was more pronounced in patients with lower HbA1c levels and insulin dose at randomization, suggesting that metabolic features may also help identify a subgroup of patients likely to respond to teplizumab [58].

Abatacept (CTLA4-Ig) has been tested in recent-onset T1D in a trial by Orban and colleagues. Patients were randomly assigned to receive abatacept (10 mg/kg) or placebo infusions. The results were promising, abatacept slowed reduction in beta-cell function for nine months in the treatment group, as evidenced by stimulated c-peptide secretion. However, the effect diminished with time; consequently, the decrease in beta-cell function became similar to that in the placebo group. Further investigation is required to determine whether the beneficial effect persists after discontinuation of abatacept [59]. Moreover, it seems to be reasonable to test the efficacy of abatacept at the early stages of T cell activation.

Another T cell co-stimulation blocking agent is alefacept, which interrupts CD2-mediated T cell co-stimulation and depletes pathogenic T cells. Therefore, it was hypothesized that alefacept might prolong the preservation of endogenous insulin secretion by the remaining beta-cells in new-onset patients by changes in effector T cells, The Immune Tolerance Network Type 1 Diabetes with Alefacept (TIDAL) study. In the TIDAL trial, participants were randomized to receive the administration of 12-week courses of alefacept (15 mg/week) or placebo over nine months. After two years post-treatment, stimulated c-peptide level was significantly higher in the treatment group than in the placebo group, insulin requirement was also lower in those receiving group alefacept, along with a reduction of hypoglycemic events by 50%. Interestingly, the younger cohort responded to alefacept more frequently than the older participants [60]. This is consistent with the experience with monoclonal CD3 antibodies, where a response was as well more significant in younger individuals. The TIDAL study has demonstrated that immunotherapy with alefacept can improve residual beta-cell function in new-onset patients and encourages further investigations to develop an immunological intervention in T1D.

Although T1D is a T cell-mediated autoimmune disease, B lymphocytes also have an impact on T1D pathogenesis through their role in antigen presentation and T cell activation. Therefore, CD20, a cell surface protein on B lymphocytes, has become a new therapeutic strategy by using rituximab, a monoclonal antibody directed against CD20. The TrialNet study group tested the efficacy of a four-dose course of rituximab (375 mg/m^2^) in recent-onset T1D patients. After a year, stimulated c-peptide levels were significantly higher in the rituximab versus placebo group [61]. Interestingly, a further two years observation has shown that rituximab delayed the reduction in c-peptide levels but did not impact insulin dose, suggesting B-cell deletion is insufficient to preserve beta-cell mass [78]. However, the actual mechanism of the effect of rituximab in T1D still is unclear and needs further investigations.

Typically, thymoglobulin or antithymocyte globulin (ATG) is used in organ transplantation; however, its potential beneficial effects had been currently investigated in T1D. Preclinical studies have shown that the combination of low-dose murine ATG plus granulocyte colony-stimulating factor (GCSF) treatment in NOD mice with recent-onset diabetes can induce disease remission. It is thought that ATG depletes pathogenic T cells while GCSF promotes regulatory T cells (Tregs). Initially, The START trial (the study of thymoglobulin to arrest type 1 diabetes) failed to show that ATG (6.5 mg/kg) monotherapy is preserving beta-cell function in new-onset T1D [62,63]. However, a pilot trial of low-dose ATG (2.5 mg/kg) and pegylated GCSF (6 mg) in individuals with T1D duration between 4–24 months suggested that low-dose ATG/GCSF preserved c-peptide. Subsequently, the Type 1 Diabetes TrialNet ATG-GCSF Study Group initiated a three-arm randomized trial (low-dose ATG/GCSF, low-dose ATG, and placebo) in new-onset patients [64]. A single course of low-dose ATG resulted in improvement in c-peptide responses and reduction of HbA1c for at least two years after T1D onset. In the low-dose ATG/GCSF group, significant preservation of c-peptide was not observed. The study group concluded that a combination of low-dose ATG and GCSF did not provide for synergistic benefit when compared with low-dose ATG monotherapy [65].

GAD-Alum was also tested in T1D stage 3 patients, again showing no preservation of endogenous insulin production. In a randomized controlled trial in patients with newly diagnosed T1D who received either GAD-Alum vaccination or subcutaneous placebo (alum alone) was no difference in fasting c-peptide concentrations or insulin requirements after 15 months [66]. Likewise, in a randomized trial of three doses of GAD-Alum, two doses with a third placebo dose (alum alone), or three placebo doses in new-onset T1D patients. One year follow-up showed no difference in stimulated c-peptide between the cohorts [79]. In a second study in which GAD-Alum or placebo was given to newly diagnosed patients, beta-cell function declined to a similar degree in both subgroups at a 15-month follow-up [67]. However, changing an administration route of GAD-Alum to intralymphatic injections might be more effective in improving beta-cell function [80,81].

The class of immunosuppressive lymphocytes known as regulatory T cells (Tregs) is critical regulators of peripheral immune tolerance. Treg deficiency is a primary cause of autoimmune and inflammatory diseases, such as T1D [82]. Indeed, disruption in the development or/and function of Tregs has been reported in T1D individuals. Then, T regulatory cell-based therapy has attracted much attention in recent times. Marek-Trzonkowska et al. carried out an autologous infusion of ex vivo expanded Tregs (single or double Tregs infusion up to the total dose of 30 × 10^6^/kg) in children with newly diagnosed T1D and compared to the untreated control group. Tregs administration was associated with an increase in Tregs number in peripheral blood and c-peptide levels. The study concluded that repetitive Tregs infusions are safe and can delay beta-cell destruction in new-onset T1D patients [68].

Similarly, Bluestone et al. have focused on the functionality of ex vivo expanded Tregs. This open-label, phase I trial enrolled 14 patients with T1D who were dived into four dosing cohorts receiving expanded ex vivo Tregs (the dose ranged from ~5 × 10^6^ to ~2.6 × 10^9^ cells in a single infusion). At one year after the transfer, around 25% of the peak level of cells remained in circulation. In comparison, c-peptide levels persisted at two years from baseline in the two cohorts receiving a lower dose of the Treg infusion [69].

Moreover, several cytokines, such as IL-2 produced by effector T cells, are necessary for the maintenance and function of Tregs in the peripheral circulation. Decreased IL-2 production has been observed in new-onset T1D patients [83]. In animal models treatment with low dose IL-2 was able to prevent T1D [84]. Therefore, cytokine-based therapy has become an attractive target for T1D prevention studies. The safety and efficacy of low dose IL-2 (aldesleuskin) were investigated by Hartemann and Bensimon et al. in T1D participants. The study showed that a dose-dependent rise in the proportion of Tregs in the intervention group compared to placebo and with no serious adverse events [70]. Another study, The Interleukin-2 Therapy of Autoimmunity in Diabetes (ITAD) in T1D youth, is currently ongoing with the primary objective of evaluating the effects of ultra-low dose IL-2 administration on residual beta-cell function in new-onset patients. The study is planned to end in 2022 [71].

Mycophenolate mofetil (MMF, Cellcept) inhibits proliferation of both T- and B-lymphocytes, therefore, it was believed that intervention with MMF could help treat T1D by stopping T cells before complete beta-cell destruction. This hypothesis was tested by the TrialNet Study Group, in which 126 new-onset patients were recruited and randomized to receive MMF, MMF plus daclizumab (an anti-interleukin IL-2 receptor monoclonal antibody that selectively binds the IL-2 receptor, inhibiting IL-2-mediated T-lymphocyte proliferation), or placebo. The results were discouraging, neither MMF alone nor in combination with daclizumab halted the destructive process of beta-cell in newly diagnosed T1D [72].

Recently, the Diabetes Virus Detection (DiViD) study has provided strong evidence for a low-grade chronic enterovirus infection and enhanced islet anti-viral responses in the pancreatic beta-cells in newly-diagnosed patients [73]. This finding has attempted to initiate the study, the Diabetes Virus Detection and Intervention Trial [DiViDInt], testing whether an intervention with anti-viral drugs may eliminate an established chronic infection of the beta-cell. The main aim of this ongoing study is to investigate the influence of a combination of two anti-viral drugs (pleconaril and ribavirin) versus placebo on the progression of T1D and residual beta-cell function in children aged 6–15 years at the onset of the disease [85]. These anti-viral drugs are attractive candidates for T1D prevention, which might be effective not only for tertiary, but also for primary and secondary prevention. Moreover, a rigorous results analysis may help our understanding of the exact mechanism of enteroviruses in the pathogeneses of T1D.

Finally, there has been a growing interest in non-insulin drugs, previously used to treat type 2 diabetes (T2D), regarding their potential protective effects on beta-cell function in autoimmune diabetes. One such oral antidiabetic agent, dipeptidyl peptidase 4 (DPP-4) inhibitors, has been shown to preserve residual beta-cell function in diabetic animal models and individuals with T2D or impaired fasting glucose (IFG) [86,87,88]. Several findings suggest that DPP-4 inhibitors may act as a regulator of the immune system by decreasing Th1 cell immune response, upregulating secretion of Th2 anti-inflammatory cytokines, and preventing the production of the pro-inflammatory cytokines [89]. Then, DPP-4 inhibitors might enhance beta-cell survival and regeneration. However, DPP-4-mediated cell signaling, and possible immune modulation are not fully clarified in autoimmune diabetes. There have been several clinical trials testing DPP-4 inhibitors alone or in combination with other drugs in autoimmune diabetes. In NOD mice, a combined intervention with DPP-4 inhibitors and proton pump inhibitors (PPIs) has increased c-peptide level and insulin secretion. Similar intervention using a combination of sitagliptin (DPP-4 inhibitor) plus lansoprazole (PPIs) has been performed in patients diagnosed with T1D within the past six months in the REPAIR-T1D study (Combination Therapy With Sitagliptin and Lansoprazole to Restore Pancreatic Beta Cell Function in Recent-Onset Type 1 Diabetes) [74]. Unfortunately, the study has shown no differences in C-peptide levels between treated vs. placebo groups. On the other hand, the DPP-IV LADA study (Protective Effects of Sitagliptin on Beta-Cell Function in Patients With Adult-onset Latent Autoimmune Diabetes) has demonstrated that one-year intervention with sitagliptin and insulin has maintained beta-cell function in LADA patients [90]. Moreover, the same study group has analyzed T-lymphocyte subsets and expression of relevant transcription factors in LADA patients treated with sitagliptin for up to 1-year. This is the first study showing that DPP-4 inhibitors might suppress the immune response by altering the frequency of CD4+ T-cell subsets on both a cellular and mRNA level in LADA patients [91]. However, the DPP-IV LADA study has had several limitations, including the small sample size, a short followed-up period, uncertainty regarding the loss of beneficial effects after treatment discontinuation, and finally, the study group included only LADA patients. Further clinical trials on large-scale will be required to confirm these findings. Currently, three ongoing large-scale trials are testing on DPP-4 inhibitors for LADA or T1D. Two of these studies are being conducted in China. One of the studies is investigating sitagliptin with insulin in children and adults with T1D [92], the second one is testing sitagliptin or/and vitamin D3 plus insulin in adult patients with LADA [93]. The third study from Norway is testing metformin plus insulin, or metformin plus sitagliptin , or/and repaglinide [94].

A tissue-specific insulin resistance (muscle, hepatic, adipose) is a common feature of T1D. In an environment of insulin resistance, pancreatic beta-cells might initiate numerous pathological programs that synergistically promote beta-cell dysfunction and apoptosis [95]. Improving peripheral insulin sensitivity in T1D could be an important component of preserving beta-cell function. The INTIMET study (Insulin Resistance in Type 1 Diabetes Managed With Metformin) is currently investigating whether metformin is effective in improving muscle, liver, and adipose insulin resistance in T1D [75]. However, many doubts remain about the oral antidiabetic agent regarding autoimmune diabetes. Even though non-insulin drugs have a positive effect on beta-cell preservation, it still might not be sufficient for the complete recovering of beta-cell mass. Despite the potential decreased insulin requirement after the intervention, T1D or LADA patients might require insulin treatment. Nevertheless, it seems reasonable to test a possible immune modulation effect of DPP-4 inhibitors at earlier T1D stages.

## 5. Discussion

Prevention of loss of beta-cell in T1D is a major goal of current diabetes research. Preventive therapies for T1D are a long-awaited not only by researchers or physicians but mostly by patients who have autoimmune diabetes. However, past studies have not provided a clear long-term beneficial effect on the preservation of beta-cell function [16,17,18,29,33,45,61]. The question then arises what the potential reason of research failure is. To date, scientists are still struggling to understand the pathogenesis of T1D, particularly the environmental determinants of T1D are only partially identified [3]. The rapid increase in T1D incidence, with a significant trend towards diagnosis at a younger age, cannot be explained by genetic shifts in such a short period. Candidate triggers should be investigated in the environment interaction. Moreover, this observation emphasizes the importance of infancy and early environmental exposures. So far, nutritional factors have been the most exanimated triggers. Unfortunately, studies (such as TRIGR, BABYDIET, NIP, or DAISY) have not proven the role of nutrients such as cow’s milk, gluten, DHA, or vitamin D in the development of T1D [14,16,17,18]. The recognition of a single factor has appeared a great challenge, as reflected by many controversial data on their significance. Studies have tended to focus on a single factor, while T1D is a heterogeneous disease with multiple different features. Several mechanisms in different pathways may eventually be responsible for the beta-cell destructive process. The disease might result from a complex interplay between multiple factors, including distinct genetic polymorphisms and environmental effects [96].

For a long time, viral infections were suspected of triggering or exacerbating the autoimmune process of the beta-cell [24,27]. However, viral vaccines or anti-viral drugs have never been tested, and now they have become attractive candidates to be studied in clinical prevention trials. In animal models, the vaccine protected mice infected with CVB from developing virus-induced T1D [28]. However, the study using the viral vaccine as preventive therapy has not been conducted in humans. The novel hexavalent vaccine, which protects against the six known CVB serotypes, is already produced. Shortly, patients will be recruited into the study. If the vaccine appears to be effective even in some participants, it will be not only clinically significant but also the role of enterovirus in the pathogenesis of T1D will be finally confirmed.

Another point is that most secondary or tertiary prevention has been the core focus for many of the previous clinical trials, even though results have not been favorable. It seems that primary prevention could be the optimal time to attempt to prevent disease onset. Modern understanding of the natural history of T1D and proposed staging has opened new opportunities to identify at-risk individuals [13]. Recent observations have demonstrated that autoantibodies can be identified in at-risk children from 6 months of age with a peak in seroconversion between 2 and 3 years [97] and children with multiple islet autoantibodies early in life have a 70% risk of diabetes within ten years and an 84% risk within 15 years [98]. The knowledge about seroconversion shortens the time of primary prevention studies using the rate of seroconversion as a trial endpoint. However, an effective screening program remains a principal challenge for T1D prevention studies. In particular, a population-wide screening program for T1D is not yet standard practice, because large-scale programs are incredibly expensive. However, there are some intervention trials, for instance, the Fr1da Insulin Intervention or the DIPP study, among high-risk for T1D participants enrolled from the general population [36,37]. Those individuals might more benefit from preventive therapies at the earliest stages since the destruction of beta-cell is not yet advanced.

Moreover, past primary prevention studies were mostly based on a modification of the susceptible environment or elimination of the exposures, which were suspected of leading to the disease onset. Again, new approaches, such as antigen-specific immunotherapy, has been tested in primary prevention studies. GPPAD-POInT is investigating oral insulin in infants who are genetically at risk for T1D [20]. Insulin was previously studied in secondary prevention trials with yet undesirable effects [31,32,33]. Perhaps, earlier immunomodulation with insulin will prevent islet autoimmunity in at-risk children. If the study succeeds, it will be an important breakthrough in the prevention of the disease.

One of the challenges of clinical trials is balancing the potential benefits against the risks. In particular, this consideration is related to introducing an agent which modulates the immune system. This therapy may have serious adverse effects. Hence, careful consideration must be a priority when choosing which agents should be tested in a prevention trial. At this point, one of the most promising types of therapy appears to be the immune therapy with an anti-CD3 monoclonal antibody, such as teplizumab. Recent studies involving patients with T1D have shown that teplizumab intervention reduces the loss of beta-cell function [40]. Prevention of T1D might become a reality in the next few years, with teplizumab likely to be the first agent commercially available to halt disease progression. Another immune-modifying agent is rituximab, which slows disease progression, but the effect was less durable compared to teplizumab [61]. Low-dose anti-thymocyte globulin, an immune system suppressant, also has delayed the loss of beta-cell function [62]. The main concern about those two agents is that they could have significant side effects, and each had only transient benefit.

Probably, the future for prevention trials of T1D will be focused mostly on teplizumab and other immune-modifying agents. Nevertheless, further studies should not only concentrate on single hypotheses or single intervention, but also the interplay between factors or agents that need to be evaluated. Primarily, it is necessary to fully understand the mechanisms of autoimmune pathogenesis and the individual response to therapies in studies participants. Alternative dosing regimens, testing agents even earlier in at-risk individuals, and a combination of various preventive therapies might be the next step in preventing T1D.

## 6. Conclusions

T1D pathogenesis is an incredibly complex process and not yet fully clarified. Still, environmental triggers of islet autoimmunity and genetic background need to be better understood. Nevertheless, improvements in understanding of the natural history of T1D, the biochemical recognition of autoantigens, identification of high-risk groups for T1D, disease prediction brought a new glimmer of hope not only for prevention, but also for reversal of T1D. Ultimately, primary prevention of islet autoimmunity will likely be the optimal approach to the prevention of T1D. Novel primary prevention clinical intervention should be started close to birth and leverage the initial seroconversion rate to shorten trials for prevention. However, screening the general population is presumably more difficult than screening relatives; therefore, mass screening for islet autoantibodies and secondary prevention could be the next choice. If beta-cell autoimmunity is developed, there is still the potential therapeutic benefit of the intervention. T1D prevention researches are developing at an exceptional rate; however, an intensive scientific effort is required to develop a sustainable method of preventing the complex autoimmune process that leads to T1D.

## Figures and Tables

**Figure 1 jcm-09-02805-f001:**
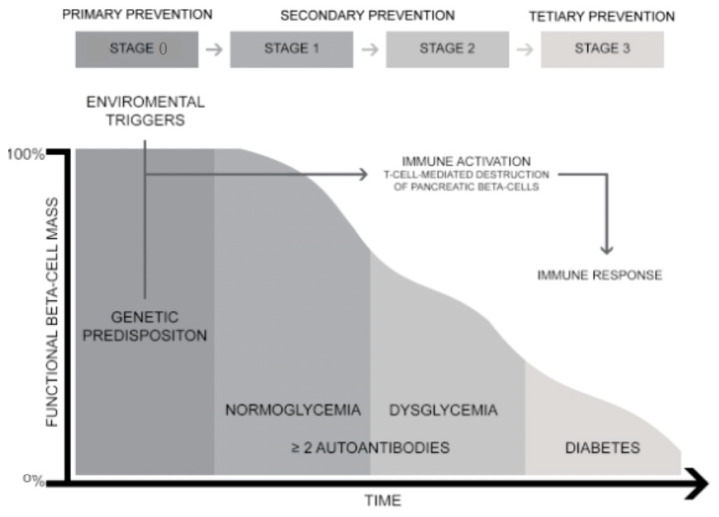
The natural history of Type 1 diabetes (T1D) is based on the referenced model [12] with recently proposed early stages of T1D [13] and prevention opportunities.

**Table 1 jcm-09-02805-t001:** Diabetes risk by human leukocyte antigen (HLA) haplotypes [2,7,8,9].

DRB1	DQB1	DQA1	Ethnic Backgrounds
**High-risk haplotypes**			
03:01	02:01	05:01	Caucasians, Koreans
04:01	03:02	03:01	Caucasians
04:02	04:02	04:01	Caucasians
04:05	03:02	03:01	Caucasians
04:01	03:03	04:05	Japanese, Koreans
03:02	03:01	08:02	Japanese
03:03	03:00	09:01	Japanese, Koreans
06:04	01:02	13:02	Japanese
**Moderate risk haplotypes**			
01	05:01	01:01	Caucasians
08:01	04:02	04:01	Caucasians
09:01	03:03	03:01	Caucasians

**Table 2 jcm-09-02805-t002:** Primary prevention trials before clinical diagnosis of T1D.

Study [Ref.]	Inclusion Criteria	Age	Intervention	Follow-Up/Primary Endpoint	Outcome	Status
TRIGR [14]	Newborns with risk-associated HLA genotypes	0–7 days	Hydrolyzed infant formula	10 years/T1D	Failed to delay or prevent the development of T1D	Completed
FINDIA [15]	Newborns with high-risk HLA	Infants	Insulin-free whey-based formula	2 years/islet autoantibodies, T1D	Reduced the incidence of autoantibodies by age 3 years	Completed
BABYDIET [16]	First degree relatives with high-risk HLA	<3 months	Gluten-free diet	3 years/islet autoantibodies	No evidence of reducing the risk for autoantibodies development	Completed
NIP [17]	Pregnant mothers and newborns with genetic risk for T1D	>24 weeks gestation/ newborns	Docosahexaenoic acid (DHA)	2 years/20% higher plasma levels of DHA	No effect on autoimmunity	Completed
DAISY [18]	First-degree relatives of patients with T1D and newborns with genetic risk	<8 years/ newborns	Vitamin D	2 years/islet autoantibodies, T1D	Failed to reduce the risk of islet autoantibodies/T1D development	Completed
Pre-POINT [19]	Individuals with familial risk	1.5–7 years	Oral insulin	3–18 months/islet autoantibodies	A high dose of daily oral insulin is safe and appears to change the immune response to insulin.	Completed
GPPAD-POInT [20]	Children with genetic risk for T1D	4–7 months	Oral insulin	7.5 years/islet autoantibodies, T1D	Not yet reported	Ongoing

**Table 3 jcm-09-02805-t003:** Secondary prevention trial in T1D.

Study	Inclusion Criteria	Age	Intervention	Follow-Up/Primary Endpoint	Outcome	Status
DENIS [29]	First-degree relatives of patients with T1D	3–12 year	Nicotinamide	3.8 years/T1D	Failed to delay of T1D development	Completed
ENDIT [30]	Family members with ICA positive but OGTT negative	<40 year	Nicotinamide	5 years/T1D	No effect on halting or preventing T1D	Completed
DPT-1 (first arm) [31]	ICA-positive T1D siblings with decreased first-phase insulin secretion	3–45 year	Parenteral insulin	5 years/T1D	The incidence of T1D in the intervention group was virtually the same as in the observation group	Completed
DPT-1 (second arm) [32]	ICA-positive T1D siblings with normal first-phase insulin secretion	3–45 year	Oral insulin	5 years/T1D	Failed to delay or prevent T1D development	Completed
TrialNet Oral Insulin [33]	Relatives with at least 2 autoantibodies	1–45 year	Oral insulin	7–8 years/T1D	No effect on reducing the risk of T1D onset	Completed
INIT [34]	Individuals with one or more T1D-related autoantibodies	4–33 year	Nasal insulin	1 year/T1D	Improved immune tolerance, but no significant effects on beta-cell function	Completed
INIT-II [35]	Relatives with at least 2 autoantibodies, HLA	4–30 year	Nasal insulin	5 years/T1D	Intranasal insulin-induced immune response, but has no effect on the prevention of T1D	Active, not recruiting
Fr1da Insulin Intervention [36]	General population with at least two islet autoantibodies	2–12 year	Oral insulin	1–7 years/T1D	Not yet reported	Ongoing
DIPP [37]	General population with high-risk genotypes for T1D	Newborn	Nasal insulin	10 years/T1D	Failed to find an effect of nasal insulin administration on T1D progression	Completed
DiAPREV-IT [38]	Individuals with multiple islet autoantibodies	4–18 year	GAD-Alum	5 years/T1D	Failed to halt the progression of the autoimmune process	Completed
DiAPREV-IT2 [39]	Individuals with multiple islet autoantibodies	10–18 year	GAD-Alum and Vitamin D	5 years/T1D	Not yet reported	Ongoing
TrialNet Teplizumab [40]	Relatives with at least 2 autoantibodies, with impaired OGTT	8–45 year	Anti-CD3 (teplizumab)	4–6 years/T1D	Teplizumab can delay T1D diagnosis a median of 2 years	Completed
TrialNet Abatacept [41]	Relatives with at least 2 autoantibodies	6–45 year	Anti–CTLA-4 (abatacept)	5–6 years/AGT, T1D	Not yet reported	Ongoing
TrialNet HCQ [42]	Individuals with multiple islet autoantibodies	3–18 year	HCQ	5–6 years/AGT, T1D	Not yet reported	Ongoing
SIMPONI [43]	Individuals with at least 2 autoantibodies, with impaired OGTT	6–21 year	TNF-a (golimumab)	5–6years/AEs	Not yet reported	Ongoing

OGTT, oral glucose tolerance test; HLA, human leukocyte antigen; HCQ, hydroxychloroquine; Aes, treatment-emergent adverse event.

**Table 4 jcm-09-02805-t004:** Tertiary prevention trials in T1D.

Study	Time from Diagnosis/ Eligibility	Age	Intervention	Follow-Up/Primary End Point	Outcome	Status
Silverstein et al. [52]	<2 weeks	4–32 year	Azathioprine and Prednisone	2.5 years/MMTT C-peptide	Short-lived remission in the treatment group	Completed
Bougneres et al. [53]	Newly diagnosed	7–15 year	Cyclosporine A	2 years/glucagon stimulation test C-peptide	Cyclosporin A transiently maintained a residual insulin secretion	Completed
Keymeulen et al. [54,55]	<4 weeks/> C-peptide 0.2 pmol/mL	12–39 year	Anti-CD3 (otelixizumab)	48 months/glucagon stimulation test C-peptide	Anti-CD3 maintained a residual beta-cell function	Completed
DEFEND-1 and -2 [56]	≤12 weeks/C-peptide 0.2–3.5 pmol/mL	12–45 year	Anti-CD3 (otelixizumab)	1 year/MMTT C-peptide	No preservation of beta-cell function	Completed
Protégé study [57]	≤12 weeks/C-peptide detectable	8–35 year	Anti-CD3 (teplizumab)	2 years/insulin dose + HbA1c MMTT C-peptide	Improved C-peptide responses in the 14-day high dose subgroup	Completed
AbATE [58]	<8 weeks	8–30 year	Anti-CD3 (teplizumab)	2 years/MMTT C-peptide	Lower insulin requirement in the treatment group	Completed
Orban et al. [59]	<14 weeks/> C-peptide 0.2 pmol/mL	6–36 year	CTLA4-Ig (abatacept)	2 years/MMTT C-peptide	Abatacept slowed the decline of beta-cell function over two years.	Completed
TIDAL [60]	<14 weeks/> C-peptide 0.2 pmol/mL *	12–35 year	Alafacept	2 years/MMTT C-peptide	C-peptide in the treatment group was significantly higher compared to placebo	Completed
TrialNet Rituximab [61]	≤12 weeks/C-peptide 0.2 pmol/mL *	8–45 year	Anti-CD20 (rituximab)	2 years/MMTT C-peptide	C-peptide levels were significantly higher in the rituximab versus the placebo group	Completed
START [62,63]	6 weeks/C-peptide * 0.4 pmol/mL	12–35 year	ATG	2 years/MMTT C-peptide	Failed to show that ATG preserves beta-cell function in new-onset T1D	Completed
TrialNet ATG-GCSF [64,65]	<14 weeks/C-peptide 0.2 pmol/mL *	12–45 year	ATG/GCSF or ATG alone	2 year/MMTT C-peptide	Low-dose ATG preserved beta-cell function and improved insulin production	Completed
Ludvigsson et al. [66]	<12 weeks/C-peptide 0.1 pmol/mL, GAD autoantibody positive	10–20 year	rhGAD65-alum	15 months/MMTT C-peptide	No difference in C-peptide concentrations or insulin requirements	Completed
TrialNet GAD Study [67]	<12 weeks/C-peptide 0.2 pmol/mL, GAD autoantibody positive	3–45 year	rhGAD65-alum	2 years/MMTT C-peptide	No change in the course of loss of insulin secretion	Completed
Marek-Trzonkowska et al. [68]	<8 weeks/C-peptide 0.1 pmol/mL	5–18 year	Infusion of ex vivo expanded Tregs	2 years/MMTT C-peptide	An increase in Tregs number in peripheral blood and c-peptide levels	Completed
Bluestone et al. [69]	>3 weeks and <24 months/C-peptide 0.1 pmol/mL *	18–45 year	Infusion of expanded polyclonal Tregs	2.5 years/MMTT C-peptide	C-peptide levels persisted out to 2+ years after transfer in several individuals	Completed
Hartemann et al. [70]	≤ 2 years	18–50 year	Aldesleukin (IL-2)	Kinetic parameters of Treg proportions variation within CD4+ T cells in peripheral blood	IL-2 induced a dose-dependent increase in the proportion of Treg cells	Completed
ITAD [71]	<6 weeks/C-peptide 0.2 pmol/mL	6–18 year	Aldesleukin (IL-2)	Differences in C-peptide over the 6 month-treatment periods between the active and placebo groups	Not yet available	Recruiting
Gottlieb et al. [72]	≤12 weeks/C-peptide 0.2 pmol/mL *	8–45 year	Mycophenolate mofetil alone or plus daclizumab	2 years/MMTT C-peptide	No effect on residual beta-cell function	Completed
DiViDInt [73]	<3 weeks	6–15 year	Pleconaril and ribavirin	3 years/MMTT C-peptide	Not yet available	Ongoing
REPAIR-T1D [74]	<6 months	11–36 year	Sitagliptin plus lansoprazole	1 year/MMTT C-peptide	No effect on residual beta-cell function	Completed
DPP-IV LADA [75]	<3 years/diagnosed with LADA	25–70 year	Sitagliptin	2 years/MMTT C-peptide	Beneficial effect on beta-cell function	Completed

* Stimulated c-peptide; MMTT, mixed meal tolerance test; HbA1c, glycated hemoglobin.

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
