# Peer review of "Prevention of Type 1 Diabetes: Past Experiences and Future Opportunities"

_jcm, 2020, doi:10.3390/jcm9092805_

Round 1

Reviewer 1 Report

The manuscript by PrzemysÅ‚aw Beik et al entitled “Prevention of type 1 diabetes: past experiences and

3 future opportunities” has been reviewed. The manuscript aims to  provide an overview of past experiences and recent findings in the prevention of T1D.

In general the manuscript is easy to read, and the outcome is of interest for the medical and scientific community.

This reviewer have only 2 suggestion.

  1. The manuscript may benefit from including a table summarizing the the HLA-DQ genotypes conferring risk for type 1 diabetes.
  2. The Fig.1. represent is the staging of type 1 diabetes proposed by Richard A Insel et al; please provide the corresponding reference. (Diabetes Care. 2015; 38(10):1964-74)

Author Response

We would like to thank the reviewer for the careful and thorough reading of this manuscript and their critical assessment of our work. We have taken the comments on board to improve and clarify the manuscript. In the following, we address their concerns point by point:

Response to Reviewer:

The first suggestion: The manuscript may benefit from including a table summarizing the HLA-DQ genotypes conferring risk for type 1 diabetes.

Reply: Thank you for this suggestion. The table summarizing diabetes risk by HLA haplotypes in populations of different ethnic backgrounds has been added to the manuscript.

The second suggestion: Fig.1. represent is the staging of type 1 diabetes proposed by Richard A Insel et al; please provide the corresponding reference. (Diabetes Care. 2015; 38(10):1964-74)

Reply: The reference has been provided.

Reviewer 2 Report

The manuscript must be reviewed, because the revision does not bring anything interesting. The conclusions are rather flat. The rationale is not well explained. There are grammatical errors throughout the manuscript. The discussion is missing. Furthermore, the review results are not scientifically relevant.  

Author Response

We would like to thank the reviewer for the careful and thorough reading of this manuscript and their critical assessment of our work. We have taken the comments on board to improve and clarify the manuscript. In the following, we address their concerns point by point:

Reply: We have asked the native speaker to revise the grammatical errors. We agree with the reviewer's opinion regarding the missing discussion. This section has been added. We hope the discussion makes the manuscript more interesting. We have tried to explain the rationale in the discussion section. Moreover, we have added some of the latest trials to make the work more relevant.

Reviewer 3 Report

This review provides a broad overview of many clinical trials for the prevention of type 1 diabetes. The content is well-written, every study is properly summarized making the manuscript easy to follow. Nevertheless, there are a few points that need to be addressed.

Major comments:

The review covers many clinical trials based on modifications of the diet and current immunotherapies, as well as those trials which are still ongoing. However, nothing is mentioned about the use of an antiviral therapy or vaccination against viruses for prevention of type 1 diabetes. For instance, the ongoing study aimed at targeting viruses in individuals at onset of disease, the Diabetes Virus Detection and Intervention Trial (DiViDIntervention). The author might need to mention this subject in the review as well.

The authors did not include any information on complementary-to-insulin pharmacological therapies such as metformin (ongoing study INTIMET), sitagliptin (ongoing study, ClinicalTrials.gov Identifier: NCT01159847) or others.

Minor comments:

Table 2:

  • The reference for the DPT-1 study, number 17 might not be correct. In the work of A.G Ziegler et al. the DPT-1 study is shortly mentioned while the complete work of Skyler et al. is cited in this review with the reference’s numbers 21 or 22.
  • The reference for the INIT-II study is not correct. The number should be 26 instead of number 2.
  • The reference for the DPT-1 study should be umber 21 instead of 17.
  • The reference for the DPT-1 second arm should be the number 22 instead of the 21.
  • The reference of the Trialnet Oral Insulin should not be the reference 23 or 24 instead of 22?

Line 190: the reference number 31 refers to the study performed in NOD mice. The reference for the study in humans is the number 32.

Line 200: Typo “in unknown” should be “is unknown”.

The authors use “while” quite often at the beginning of sentences and it is not always right. Most of these could be deleted without changing the meaning.

Author Response

We would like to thank the reviewer for the careful and thorough reading of this manuscript and their critical assessment of our work. We have taken the comments on board to improve and clarify the manuscript. In the following, we address their concerns point by point:

Response to Reviewer 3:
Major comments:

The review covers many clinical trials based on modifications of the diet and current immunotherapies, as well as those trials which are still ongoing. However, nothing is mentioned about the use of an antiviral therapy or vaccination against viruses for prevention of type 1 diabetes. For instance, the ongoing study aimed at targeting viruses in individuals at onset of disease, the Diabetes Virus Detection and Intervention Trial (DiViDIntervention). The author might need to mention this subject in the review as well.

Reply: The trials regarding a vaccination targeting enteroviruses and antiviral therapy for prevention of type 1 diabetes have been added. Please see: Page 4, line 114 (vaccination); Page 13, line 399 (antiviral agents)

The authors did not include any information on complementary-to-insulin pharmacological therapies such as metformin (ongoing study INTIMET), sitagliptin (ongoing study, ClinicalTrials.gov Identifier: NCT01159847) or others.

Reply: The mentioned studies have been included. Please see: Page 14, line 442 (INTIME); Page 13, 410 line (sitagliptin studies).

Minor comments:

Table 2:
The reference for the DPT-1 study, number 17 might not be correct. In the work of A.G Ziegler et al. the DPT-1 study is shortly mentioned while the complete work of Skyler et al. is cited in this review with the reference’s numbers 21 or 22.
Reply: The reference has been corrected.

The reference for the INIT-II study is not correct. The number should be 26 instead of number 2.
Reply: The reference has been corrected.

The reference for the DPT-1 study should be umber 21 instead of 17.
Reply: The reference has been corrected.

The reference for the DPT-1 second arm should be the number 22 instead of the 21.
Reply: The reference has been corrected.

The reference of the Trialnet Oral Insulin should not be the reference 23 or 24 instead of 22?
Reply: The reference has been corrected.

Line 190: the reference number 31 refers to the study performed in NOD mice. The reference for the study in humans is the number 32.
Reply: The reference has been corrected.
Line 200: Typo “in unknown” should be “is unknown”.
Reply: Done

The authors use “while” quite often at the beginning of sentences and it is not always right. Most of these could be deleted without changing the meaning.
Reply: Done

Reviewer 4 Report

Sir,

Thank you for offering me the opportunity to review this article for Journal of Clinical Medicine.

This is a narrative review on prevention of type 1 diabetes mellitus. The article is interest and the authors did a lot of work.

A drawback is that there are two recent very comprehensive reviews on the topic published: Changing the landscape for type 1 diabetes: the first step to prevention. Lancet 2019 Oct 5;394(10205):1286-1296. doi: 10.1016/S0140 6736(19)32127-0. Epub 2019 Sep 15); and Diabetologia. Genetics, pathogenesis and clinical interventions in type 1 diabetes. 2017 Aug;60(8):1370-1381. doi: 10.1007/s00125-017-4308-1. There is also an older one in Nature (Nature. 2010 Apr 29;464(7293):1293-300. doi: 10.1038/nature08933).

Other comments:

1. Line 32: “islet-cell antibodies (ICA)”; these are also autoantibodies like anti-GAD, anti-IA2 etc.
2. Previous studies define pre-stage 1 as stage 0 in the pathogenesis of the disease; it is better to follow a standard staging system to describe the pathogenetic evolution of T1DM.

Author Response

We would like to thank the reviewer for the careful and thorough reading of this manuscript and their critical assessment of our work. We have taken the comments on board to improve and clarify the manuscript. In the following, we address their concerns point by point:

Response to Reviewer:

A drawback is that there are two recent very comprehensive reviews on the topic published: Changing the landscape for type 1 diabetes: the first step to prevention. Lancet 2019 Oct 5;394(10205):1286-1296. doi: 10.1016/S0140 6736(19)32127-0. Epub 2019 Sep 15); and Diabetologia. Genetics, pathogenesis and clinical interventions in type 1 diabetes. 2017 Aug;60(8):1370-1381. doi: 10.1007/s00125-017-4308-1. There is also an older one in Nature (Nature. 2010 Apr 29;464(7293):1293-300. doi: 10.1038/nature08933).

Reply: We agree with the reviewer's opinion. Recently, there are three excellent papers published on type 1 diabetes prevention. We have carefully read the mentioned reviews. The clinical trials discussed by us, at some point, overlap with those studies described by the authors of published papers. The main goal of the submitted review was to give a general overview of past experiences and recent findings in the prevention of type 1 diabetes. Therefore, we reviewed the most significant completed studies, which are also mentioned by other authors. However, in our work, we also presented the most recent clinical trials from the end of 2019 and the beginning of 2020, which are not listed in indicated papers. The field of type 1 diabetes prevention is changing rapidly, since defining stages of the disease. We believe our review gives a border overview on this topic because we also detailed some completed studies not discussed by those authors. Then, our paper might bring a new perspective.

Other comments:

  1. Line 32: “islet-cell antibodies (ICA)”; these are also autoantibodies like anti-GAD, anti-IA2 etc.

Reply: The authors has listed 5 autoantibodies (ICA, GAD-ab, IA2-ab, IAA, ZnT8) related to type 1 diabetes. Please see page: 1, line 32 to 35.

  1. Previous studies define pre-stage 1 as stage 0 in the pathogenesis of the disease; it is better to follow a standard staging system to describe the pathogenetic evolution of T1DM

Reply: According to the suggestion, we have changed a staging system.

Round 2

Reviewer 2 Report

The research work carried out by the authors is very interesting and innovative. The research is well developed, but conclusions are missing, so the manuscript is acceptable for publication.